# Two Cases of Pancytopenia with Coombs-Negative Hemolytic Anemia after Chimeric Antigen Receptor T-Cell Therapy

**DOI:** 10.3390/ijms22115449

**Published:** 2021-05-21

**Authors:** Dominik Kiem, Michael Leisch, Daniel Neureiter, Theresa Haslauer, Alexander Egle, Thomas Melchardt, Max S. Topp, Richard Greil

**Affiliations:** 1Oncologic Center, Department of Internal Medicine III with Haematology, Medical Oncology, Haemostaseology, Infectiology and Rheumatology, Paracelsus Medical University, 5020 Salzburg, Austria; d.kiem@salk.at (D.K.); m.leisch@salk.at (M.L.); a.egle@salk.at (A.E.); t.melchardt@salk.at (T.M.); 2Cancer Cluster Salzburg, 5020 Salzburg, Austria; 3Laboratory for Immunological and Molecular Cancer Research (SCRI-LIMCR), Salzburg Cancer Research Institute, 5020 Salzburg, Austria; th.haslauer@salk.at; 4Institute of Pathology, Paracelsus Medical University Salzburg, 5020 Salzburg, Austria; d.neureiter@salk.at; 5Medical Clinic and Polyclinic II, University Hospital Würzburg, 97080 Würzburg, Germany; topp_m@ukw.de

**Keywords:** CAR T-cell, hemolytic anemia, prolonged cytopenia

## Abstract

Background: Chimeric antigen receptor (CAR) T-cells are changing the therapeutic landscape of hematologic malignancies. Severe side effects include cytokine release syndrome (CRS) and immune effector cell-associated neurotoxicity syndrome (ICANS), but prolonged cytopenia has also been reported. The underlying mechanism for prolonged cytopenia is poorly understood so far. Cases: Severe pancytopenia with grade 2-3 anemia was marked 2–3 months after treatment. Laboratory evaluation revealed undetectable levels of haptoglobin with increased reticulocyte counts. Coomb’s tests were negative, no schistocytes were detected on blood smear, and infectious causes were ruled out. Increased erythropoiesis without lymphoma infiltration was noted on bone marrow biopsy. A spontaneous increase in haptoglobin and hemoglobin levels was observed after several weeks. For one patient, peripheral CAR-T levels were monitored over time. We observed a decline at the same time as hemoglobin levels began to rise, implying a potential causality. Conclusion: To our knowledge, we describe the first two cases of Coombs-negative hemolytic anemia after CAR-T treatment for B-cell lymphoma. We encourage routine monitoring for hemolytic anemia after CAR-T treatment and also encourage further investigations on the underlying mechanism.

## 1. Introduction

Chimeric antigen receptor (CAR) T-cells have revolutionized the therapeutic landscape of hematologic malignancies. Since 2017, the CD19 CAR-T cell products tisagenlecleucel and axicabtagen-ciloleucel/KTE-C19 have been approved for relapsed or refractory diffuse large B-cell lymphoma (DLBCL) and pediatric acute B-lymphoblastic leukemia (B-ALL) [1,2]. Brexucabtagene autoleucel (KTE-X19) is approved for the treatment of relapsed refractory mantle cell lymphoma [3]. CAR-T cells are also under investigation in other types of solid and hematologic malignancies [4].

Despite encouraging efficacy, CAR-T treatment is associated with relevant side-effects. Most commonly observed are cytokine release syndrome (CRS) and immune effector cell-associated neurotoxicity syndrome (ICANS). Cytopenia typically occurs soon after lymphodepleting chemotherapy as a result of bone marrow toxicity [5]. However, prolonged (i.e., lasting longer than 28 days) grade 3-4 cytopenias are reported in 32% and 38% of patients in the ZUMA-1 [6] and JULIET [7] trials, respectively, and 26% of patients in the ZUMA-2 [3] trial had grade 3-4 cytopenias after 90 days (Table 1) [5]. In these cases, bone marrow biopsy may show severe hypocellularity [8,9]. Others, however, have also described normal bone marrow cellularity [5,10], as well as cytopenia, even in the absence of induction chemotherapy [11], suggesting a mechanism other than chemotherapy-induced myelosuppression. Current literature reports an association of prolonged cytopenia with CRS or ICANS > grade 3, as well as baseline cytopenia, and inflammatory markers [12].

Hereinafter, we report on two patients who developed severe delayed pancytopenia with self-limited Coombs-negative hemolytic anemia after CAR T-cell therapy.

## 2. Results

### 2.1. Case 1

Patient A is a 51-year-old male diagnosed with DLBCL, Ann Arbor stage IV EB, in October 2018 without bone marrow infiltration. After treatment with rituximab, cyclophosphamide, doxorubicin, vincristine, and prednisone (R-CHOP), the patient achieved a partial remission (PR) and he received consolidative radiotherapy. After progression, the patient was treated with rituximab, gemcitabine, dexamethasone, and cisplatin (R-GDP), followed by high-dose chemotherapy with autologous stem cell support. He had an early disease progression two months thereafter and received CAR T-treatment (tisagenlecleucel) as part of a clinical trial. Bridging therapy with bendamustine, rituximab, and polatuzumab was followed by lymphodepletion with fludarabine and cyclophosphamide and CAR T-cell therapy in April 2020. Initially, the patient developed grade 2 cytokine release syndrome that required treatment with tocilizumab. One month after CAR-T treatment, complete remission (CR) was seen on PET scans and there were no CD19 + cells detectable in the peripheral blood. At that time, marked pancytopenia was observed, which further aggravated between day 65 and 90, with grade 2 macrocytic anemia, grade 3 thrombocytopenia, grade 3 leukopenia, and grade 4 neutropenia (Figure 1). Nadir values were 8.3 mg/dL for hemoglobin, 15 × 10^9^/L for platelets, 0.33 × 10^9^/L for leukocytes, and 0 × 10^9^/L for neutrophils. The patient required transfusion of erythrocytes on day 7 and platelets on day 85. Evaluation of the cytopenia was performed and revealed undetectable levels of haptoglobin and elevated free serum hemoglobin, with elevated reticulocyte counts and normal bilirubin values (Appendix A). Mild lactate dehydrogenase (LDH) elevation had been observed since day 27 after treatment and peaked on day 85. Coombs test was negative and no schistocytes were detected in the peripheral blood smear. PCRs for EBV and CMV were repeatedly negative and the patient showed no clinical signs of infection. The patient was started on epoetin zeta and filgrastim without improvement of blood cell counts. Bone marrow biopsy ruled out lymphoma infiltration and showed massively hyperplastic erythropoiesis with hypoplastic megakaryo- and granulopoiesis (Figure 2). Staining for CD19 (for B-cells) and CD3 (for T-cells, including CAR-T) did not show overlap (Appendix A). In addition to that, B-cell aplasia with hypogammaglobulinemia was observed. Ferritin levels were constantly elevated between 1400 and 2200 µg/dL after CAR T-cell treatment and CRP was normal to minimally elevated up to 1.2 mg/dL. After day 114, leukocytes and hemoglobin values started to increase spontaneously and growth factors were discontinued. Since then, the anemia has remained stable at grade 1, haptoglobin levels have returned to normal (Figure 1 and Appendix A) and the patient is in ongoing remission.

### 2.2. Case 2

Patient B is a 57-year-old woman diagnosed with mantle cell lymphoma Ann Arbor IV AE presenting with massive splenomegaly in September 2019. She was classified as high risk according to MIPI and had a complex karyotype with a TP53 mutation. The patient was refractory to initial treatment with R-CHOP alternating with rituximab, dexamethasone, cytarabine, and cisplatin (R-DHAP). A therapy with ibrutinib and venetoclax was initiated and partial remission with minimal residual disease (MRD) positivity was achieved. Given the genetic high risk features and chemotherapy refractoriness, the patient was evaluated for CAR-T cell therapy at the university hospital Wuerzburg (Germany,) and treatment with Brexucabtagene autoleucel (KTE-X19) in an early access program was completed in June 2020. She developed CRS grade 2 and neurotoxicity grade 3 which required treatment with tocilizumab and steroids. MRD negativity was detected on day 30 after treatment. Subsequently, she developed progressively worsening pancytopenia starting on day 37 (lowest point between day 51 and 59) with grade 3 (macrocytic) anemia, grade 4 thrombocytopenia, and grade 4 leuko- and neutropenia requiring filgrastim (Figure 1). For this patient, the nadir values were 7.7 mg/dL for hemoglobin, 24 × 10^9^/L for platelets, 0.67 × 10^9^/L for leukocytes and 0.1 × 10^9^/L for neutrophils. On day 58, anemia required red blood cell transfusion. Lab tests showed undetectable levels of haptoglobin, slightly increased bilirubin, and increased LDH (Appendix A). The reticulocyte counts were mildly elevated with a maximum on day 65. No schistocytes were seen on blood smear. Coombs test as well as PCRs for EBV, CMV, and parvovirus were negative and the patient had no clinical signs of infection. Bone marrow biopsy showed again massively increased erythropoiesis with hypoplastic granulopoiesis (Figure 2). The patient had total CD19 B-cell aplasia with secondary hypogammaglobulinemia. Furthermore, in this patient, ferritin levels were elevated (between 1000 and 1700 µg/dL) after CAR T-cell therapy with normal to slightly elevated CRP up to 1.1 mg/dL. A constant increase of thrombocyte levels starting from day 65 was noted and thrombocytopenia improved to grade 1 by day 121. Hemoglobin levels finally started to increase spontaneously after day 81 and normalized by day 121. Haptoglobin levels normalized by day 121. Neutrophil counts turned within reference levels on day 142 under continued growth factor support. For this patient, we had the possibility to measure CAR-T cell levels in peripheral blood samples by flow cytometry at our research laboratories. We noticed a sharp decline of CAR-T cell levels in the peripheral blood by day 121 paralleled by normalization of haptoglobin levels (Figure 1 and Figure 3) and reconstitution of CD19 positive B-cells. We attempted to assess possible interaction within the bone marrow, but staining of CD19 (for B-cells) and CD3 (for T-cells, including CAR-T) did not show overlap (Appendix A). The same was true for patient A. At this point, the patient is still in CR.

## 3. Discussion

These two cases are examples of severe, prolonged, but self-limited pancytopenia with features of hemolytic anemia after CAR-T cell therapy. In both cases, we observed profound cytopenia 2–3 months after treatment, which was further evaluated. At our center, nine patients have been treated with CAR-T cells so far, but the phenomenon of hemolytic anemia was not observed in any of the other patients.

We observed pancytopenia on complete blood counts with a decline in hemoglobin levels. The combination with undetectable levels of haptoglobin and slight reticulocytosis led us to the diagnosis of hemolytic anemia with negative Coombs test. As a next step, we looked for common causes of hemolytic anemia (i.e., microangiopathic hemolytic anemia, viral infections, medications) [14] and did not find any underlying etiology. When we evaluated the bone marrow, we were able to rule out lymphoma infiltration of the bone marrow as the underlying cause for the pancytopenia. We observed an increase in erythropoietic precursors, while thrombo- and granulopoiesis were decreased. Of note, bone marrow did not show any signs for hemophagocytosis, and patients did not fulfill HLH-2004 diagnostic criteria for hemophagocytic lymphohistiocytosis [15]. Since we did not find CD19 expression in erythropoietic precursors, we ruled out an on-target off-tumor effect.

Decreased bone marrow cellularity is commonly described after CAR-T treatment, and the reduction in thrombo- and granuopoiesis is in line with previous reports about prolonged cytopenias after CAR-T treatment [5,8,9,10]. However, erythropoietic precursors were increased in our cases, which is in accordance with the observed peripheral destruction of RBCs. In one patient, we had the possibility to measure CAR-T levels in the peripheral blood at this time. Interestingly, we observed a decline in peripheral CART- cell levels over time coinciding with resolution of hemolysis. Given the timely correlation with the CAR-T treatment, we speculated whether there was an association with the hemolytic anemia.

Probably, destruction of RBCs was caused by cytotoxic granules, which are released upon CAR-T cell binding to the target cell as a sort of “bystander effect”, but we can only speculate about this. We argue that most of CAR-T cell-mediated lymphoma cell destruction will take place at the tumor site shortly after infusion (in our case lymph nodes, spleen, liver, and bone marrow) and not the peripheral blood. When patients start to reconstitute some CD19 positive healthy B-cells in the blood after some weeks to months, they will eventually be recognized by CAR-T cells and consecutively be eliminated. Probably, some RBCs will get destroyed as a part of this process. In line with this postulated mechanism, we observed an incline in CD19 levels exactly during the time of hemolysis and declining CAR-T cells.

Although the observed changes in both cases seem very similar, it is noteworthy that the patients were treated with different CAR-T constructs, which defer in their costimulatory domain and persistence. As such, persistence of tisagenlecleucel in the peripheral blood has been reported for up to two years for patients in CR in the Juliet trial [7], with no obvious correlation with clinical outcomes. Similarly, the Zuma-2 trial describes persistence of brexucabtagene autocel for up to two years [3]. Still, for almost one-fifth of patients with ongoing response, no CAR-T cells were detectable six months after treatment. For both constructs, there seems to be a large variability in persistence of CAR-T cells between patients treated with the same product, and cellular kinetics will most likely differ between different products. Therefore, persistence of CAR-T cells in the individual patient should be considered an important factor when assessing causes for pancytopenia.

We want to point out important limitations of our case study, since we did not look for signs of complement activation (i.e., C3 and C4 levels) and did not test for paroxysmal nocturnal hemoglobulinuria as possible differential diagnosis. We would recommend testing for this if future similar cases occur.

Guidelines for management of CAR-T related cytopenias are regularly updated in a fast evolving field. As reviewed by Penack and Koenecke in 2020, prolonged cytopenia is still of unclear origin in most patients [16]. Regarding management recommendations, Neelapu lists the possibility of growth factor support and transfusions as necessary options and advice antibiotic/antimycrobial prophylaxis for prolonged neutropenia [8]. The current EBMT guidelines recommend routine monitoring of complete blood count 28 days after therapy but do not specifically advise on monitoring for hemolysis and do not give specific recommendations regarding management [17].

## 4. Conclusions

To our knowledge, this is the first report of two cases with Coombs-negative hemolytic anemia following CAR-T cell treatment, however, the underlying mechanism remains elusive at this point and requires further investigations. Given these observations, we suggest regular monitoring for hemolytic anemia following CAR-T treatment in routine practice as well as in clinical trials.

## Figures and Tables

**Figure 1 ijms-22-05449-f001:**
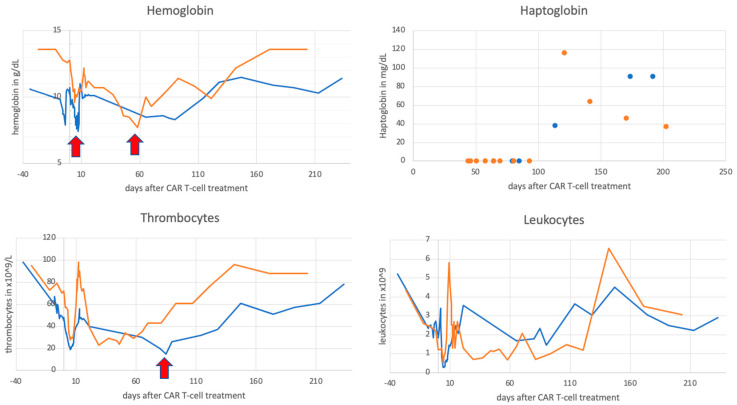
Course of hemoglobin, haptoglobin, leukocytes, thrombocytes over time. Blue = patient A, orange = patient B, arrows indicate blood transfusions (erythrocytes and thrombocytes, respectively).

**Figure 2 ijms-22-05449-f002:**
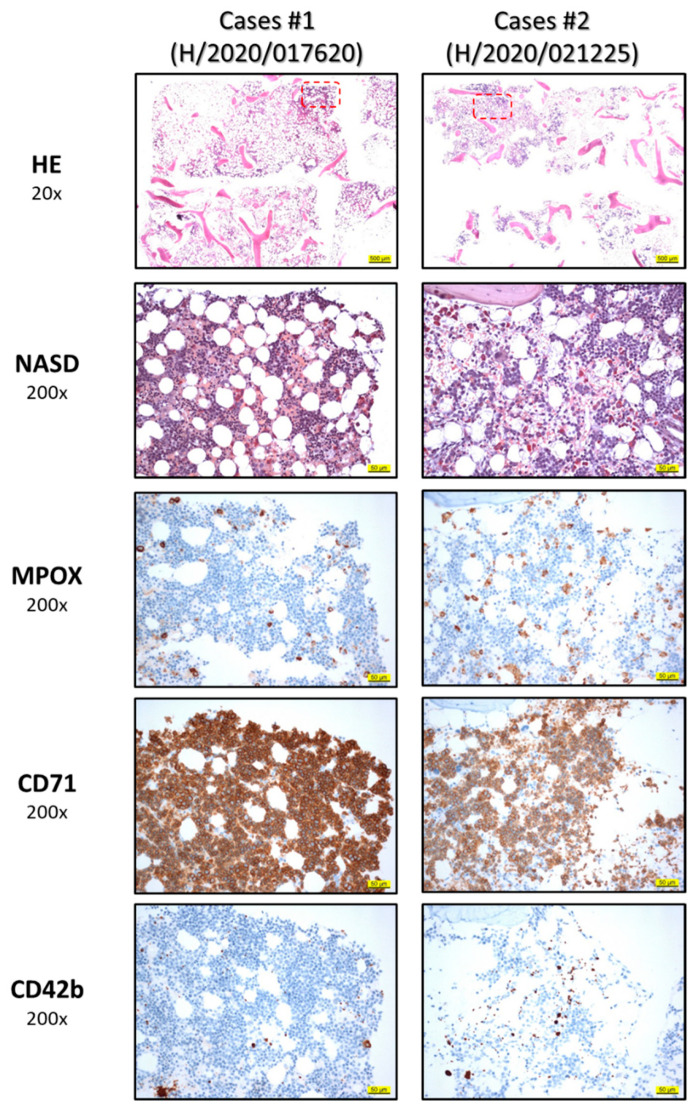
Patient A: The HE-staining in combination with NASD-reaction of the bone marrow demonstrated a slightly increased cellularity with increased erythropoesis (CD71) as well as reduced granulopoiese (MPOX) and megakaryopoisis (CD42b). The additional imunhistochemicals (E-Cadherin, CD14 and CD68, not shown) revealed distinct dysplastic changes of the hematopoesis. Apart from that, low-grade bone marrow fibrosis with discrete hemosiderin deposition were found (not shown). Patient B: The routine stainings (HE-staining and NASD-reaction) revealed an enhanced cellularity of the hematopoesis in comparison to Patient A. Again, the erythropoiesis (CD71) was clearly increased, whereby the granulopoiesis (MPOX) was decreased with left shift and the megacaryopoeisis (CD42b) was normocellular. Additionally, hemosiderin deposition was increased (not shown).

**Figure 3 ijms-22-05449-f003:**
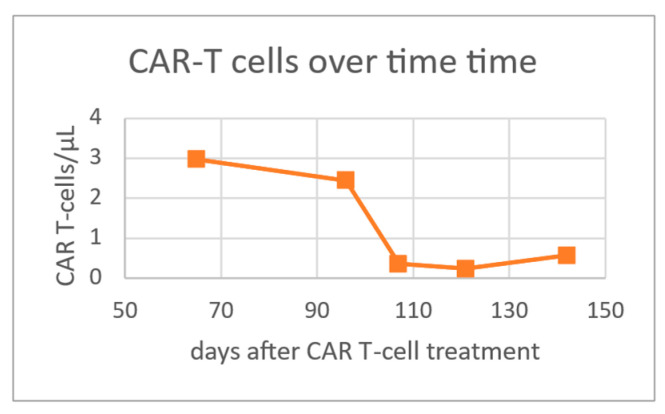
Course of CAR-T cell levels over time.

**Table 1 ijms-22-05449-t001:** Occurrence of late cytopenias in pivotal clinical trials.

Product	Trial	Prolonged Cytopenia	
		All Grades	Grade 3/4
Tisagenlecleucel	Eliana [13]	37% (day 28)	32% (day 28)
Tisagenlecleucel	Juliet [7]	44% (day 28)	32% (day 28)
Axicabtagene ciloleucel	Zuma-1 [6]	55% (day 30)	38% (day 30)
Brexucabtagene autoleucel	Zuma-2 [3]	N/A	26% (day 90)

## Data Availability

Data can be obtained from the corresponding author upon reasonable request.

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
