# Peer review of "Two Cases of Pancytopenia with Coombs-Negative Hemolytic Anemia after Chimeric Antigen Receptor T-Cell Therapy"

_ijms, 2021, doi:10.3390/ijms22115449_

Round 1

Reviewer 1 Report

Comments to the authors

Kiem et al. report two patients that developed pancytopenia and Coombs-negative hemolytic anemia after CAR-T cell therapy. Authors have excluded HLH/MAS and lymphoma relapse in BM as a cause of observed pancytopenia. Based on low haptoglobin level and increased reticulocytosis in PB as well as increased erythropoiesis in BM authors conclude that anemia results from Coombs-negative hemolysis. However, the mechanism of pancytopenia and hemolysis remain to be elucidated although authors present data that shows decline of the CAR-T cells for second patient at the same time as the pancytopenia alleviated.

CAR-T cell therapy is still relatively novel treatment and reports describing new relevant safety issues and toxicities are important for hematologist and oncologists. I think that this case report may be publishable in International Journal of Molecular Sciences if the manuscript is revised according to following comments:

  1. Did patients receive red cell or platelet infusions? Probably not, but this is clinically relevant and should be clearly stated.
  2. For patient B blood counts starting from D+37 after CAR-T infusion are presented. It is necessary to show also CBC values before CAR-T infusion and during the first month of follow-up after the infusion. Patient B received intensive chemotherapy and other drugs known to induce cytopenia (rituximab and ibrutinib) before CAR-T cell infusion. Thus it is extremely relevant to describe the situation before the CAR-T cell treatment. Patient had significant acute side effects after CAR-T infusion (CRS grade 2 and grade 3 neurotoxicity) and received both tocilizumab and steroids. Accordingly, it is important to describe changes in CBC during the acute follow-up in order to evaluate cytopenias described in the manuscript.
  3. Patients received different CAR-T constructs. Patient A was treated with tisagenlecleucel by Novartis and patient B axicabtagene-ciloleucel by Kite Pharma. Although these CAR-T constructs have the same target (CD19), their other qualities differ. Importantly, there may be difference in persistence of the constructs in the patients. This should be discussed as the data on CAR-T numbers are presented only from patient B and the etiology of pancytopenia is proposed to relate with the level of CAR-T cells in the patient.
  4. In Figure 1. scales on y-axes in panels depicting Hb concentration and platelet count make results very difficult to read. I suggest y-axis scale from 5 to 15 g/dl for Hb and 0 to 100 x 109/L for platelets.
  5. In Figure 1 and in supplemental material (both table and figure) peculiar units for cell concentrations are used. I suggest using common units, e.g. x 109/L for leukocytes and platelets.
  6. Unit for relative reticulocyte count should be % instead of T/L in Supplementary figure 1.
  7. If lab values of the patients are presented only in Supplements, I think that at least nadir values of important CBC values (i.e. leukocyte count, neutrophil count, platelet count, and Hb) should be included also in the Results section to ease the reading of the article.
  8. I don’t see what relevant additional information are provided by showing both absolute and relative number of CAR-T cells in patient B (Figure 3). I think that only absolute numbers would be adequate.
  9. 4 may be presented in Supplements as it presents negative results and doesn’t bring major contribution to the discussion about the mechanism of the pancytopenia.
  10. Authors have excluded HLH/MAS as an etiology of pancytopenia by measuring ferritin and by negative finding of hemophagocytosis in bone marrow examination. Did authors measure cytokines or soluble IL-2 receptor to investigate activity of the immune system?
  11. To me it seems that there are two different processes in the patients. First, they have Coombs-negative hemolytic anemia (increased reticulocytes and erythropoiesis, high LD, elevated bilirubin and low haptoglobin). Second, they have leuko/neutropenia and thrombocytopenia due to suppressed production in BM. What could be common reason for these phenomena? Or are there two different processes that start and alleviate simultaneously (that seems to me quite unlikely)? Any signs of complement activation? Authors should discuss this problem more profoundly.
  12. Authors suggest that release of cytotoxic granules (presumably by CAR-T cells) would cause hemolysis as “bystander effect”. To me this explanation seems illogical as patients have reached remission and also the amount of the target of CAR-T cells (CD19) has significantly diminished at the time when hemolysis was detected. Why this “bystander effect” does not manifest earlier when CAR-T activity is at its highest and CAR-T cells are releasing cytotoxic granules most?
  13. There is a typo in line 37. “Pediatric B-acute lymphoblastic anemia” should be “pediatric acute B-cell lymphoblastic leukemia”.

Author Response

Dear reviewer,

We would like to thank for the constructive feedback and are happy to implement the following points as described below. We believe that our manuscript has gained quality due to these modifications. Due to multiple important comments, we found ways to restructure our discussion significantly and believe that this has greatly improved the quality of our work. 

  1. Did patients receive red cell or platelet infusions? Probably not, but this is clinically relevant and should be clearly stated.

Response: Yes, both patients received RBCs on one occasion and patient B required an additional transfusion of platelets. This has been inserted in lines 75-76 (“The patient required transfusion of erythrocytes on day 7 and platelets on day 85.”) and in line 124-125 (“On day 58, anemia required red blood cell transfusion.”). In addition to that, arrows were inserted into figure 1 to highlight transfusions. (Line 103-106)

  1. For patient B blood counts starting from D+37 after CAR-T infusion are presented. It is necessary to show also CBC values before CAR-T infusion and during the first month of follow-up after the infusion. Patient B received intensive chemotherapy and other drugs known to induce cytopenia (rituximab and ibrutinib) before CAR-T cell infusion. Thus it is extremely relevant to describe the situation before the CAR-T cell treatment. Patient had significant acute side effects after CAR-T infusion (CRS grade 2 and grade 3 neurotoxicity) and received both tocilizumab and steroids. Accordingly, it is important to describe changes in CBC during the acute follow-up in order to evaluate cytopenias described in the manuscript.

Response: We thank the reviewer for this valuable input. Patient B was not treated at our center and was first seen on D+37. We have contacted the treating center and included blood values from the time the patient was seen the last time before CAR-T treatment at our center and added available blood results from D0 to D+37.  Also, additional lab values are added for the patient treated at our center in the month following to CAR T-cell therapy are included. In addition to that, we included the CBC during the month before CAR T-treatment (figure 1, line 103-106 and supplementary figure and supplementary tables 1+2)

  1. Patients received different CAR-T constructs. Patient A was treated with tisagenlecleucel by Novartis and patient B axicabtagene-ciloleucel by Kite Pharma. Although these CAR-T constructs have the same target (CD19), their other qualities differ. Importantly, there may be difference in persistence of the constructs in the patients. This should be discussed as the data on CAR-T numbers are presented only from patient B and the etiology of pancytopenia is proposed to relate with the level of CAR-T cells in the patient.

Response: We agree with the reviewer in this regard and have added clinical trial data on CAR-T persistence for both products to the discussion. We noticed in the process of the revision, that we falsely used the term Axicabtagene ciloleucel when in fact the patient was treated with brexucabtagene autoleucel (KTE-X19). We have made the appropriate corrections (line 117).

“Although the observed changes in both cases seem very similar, it is noteworthy that the patients were treated with different CAR-T constructs, which defer in their costim-ulatory domain and persistence. As such, persistence of tisagenlecleucel in the peripheral blood has been reported for up to two years for patients in CR in the Juliet trial with no obvious correlation with clinical outcomes. Similarly, the Zuma-2 trial, describes persistence of brexucabtagene autocel for up to two years. Still, for almost one fifth of patients with ongoing response, no CAR T-cells were detectable six months after treatment. For both constructs, there seems to be a large variability in persistence of CAR T-cells between patients treated with the same product, and cellular kinetics will most likely differ be-tween different products. Therefore, persistence of CAR T-cells in the individual patient should be considered an important factor when assessing causes for pancytopenia.” (line 203-227)

  1. In Figure 1. scales on y-axes in panels depicting Hb concentration and platelet count make results very difficult to read. I suggest y-axis scale from 5 to 15 g/dl for Hb and 0 to 100 x 109/L for platelets.

Response: We agree and have made the required changes (Figure 1, line 91)

  1. In Figure 1 and in supplemental material (both table and figure) peculiar units for cell concentrations are used. I suggest using common units, e.g. x 109/L for leukocytes and platelets.

Response: We agree with the reviewer and have made the suggested adjustments have been made. (Figure 1, 103-104)

  1. Unit for relative reticulocyte count should be % instead of T/L in Supplementary figure 1.

Response: We made the suggested changes (Supplementary Figure 1). The figure, however, is not included in the main manuscript.

  1. If lab values of the patients are presented only in Supplements, I think that at least nadir values of important CBC values (i.e. leukocyte count, neutrophil count, platelet count, and Hb) should be included also in the Results section to ease the reading of the article.

Response: We thank the reviewer for this valuable input, CBC nadir values for patient A have been inserted in line 73-75 (“Nadir values were 8,3 mg/dL for hemoglobin, 15 x109/L for platelets, 0,33 x109/L for leukocytes and 0 x109/L for neutrophils”) and in line 123-124 for patient B (“For this patient, the nadir values were 7,7 mg/dL for hemoglobin, 24 x109/L for platelets, 0,67 x109/L for leukocytes and 0,1 x109/L for neutrophils.”).

  1. I don’t see what relevant additional information are provided by showing both absolute and relative number of CAR-T cells in patient B (Figure 3). I think that only absolute numbers would be adequate.

Response: The relative CAR T-cell levels have been eliminated from the figure. (line 165)

  1. 4 may be presented in Supplements as it presents negative results and doesn’t bring major contribution to the discussion about the mechanism of the pancytopenia.

Response: This is a valid point, Figure 4 is now shown as supplementary figure 2. The figure was eliminated from line 167.

  1. Authors have excluded HLH/MAS as an etiology of pancytopenia by measuring ferritin and by negative finding of hemophagocytosis in bone marrow examination. Did authors measure cytokines or soluble IL-2 receptor to investigate activity of the immune system?

Response: We thank the reviewer for this important note. Cytokine- and soluble IL-2 receptor levels are not routinely available in our central laboratory and have not been assessed because we had no clinical suspicion for HLH/MAS in the absence of fever, elevated triglycerides, rising ferritin or hemophagocytosis on bone marrow smears. Rather, we discussed this option only when preparing this case report. We argue, however, that even an increase in soluble IL-2 receptor would not be sufficient for our patients to fulfill enough criteria of the HLH-2004 diagnostic criteria (Please also refer to PMC7923749, kindly suggested by reviewer 2). “Of note, bone marrow did not show any signs for hemophagocytosis and patients did not fulfill HLH-2004 diagnostic criteria for hemophagocytic lymphohistiocytosis.” (line 180-182)

  1. To me it seems that there are two different processes in the patients. First, they have Coombs-negative hemolytic anemia (increased reticulocytes and erythropoiesis, high LD, elevated bilirubin and low haptoglobin). Second, they have leuko/neutropenia and thrombocytopenia due to suppressed production in BM. What could be common reason for these phenomena? Or are there two different processes that start and alleviate simultaneously (that seems to me quite unlikely)? Any signs of complement activation? Authors should discuss this problem more profoundly.

Response: We agree with the reviewer, that it is hard to clearly define a common reason for the cytopenias, as we have tried to formulate in the discussion. We think that the lab results are in line with hemolytic anemia (increased erythropoiesis, low haptoglobin ecc.). The reason for decreased platelet- and leukocyte counts seems different (i.e. not a peripheral destruction like in ITP or fragmentation syndromes). It rather seems that there is a decreased production in the bone marrow (decreased granulo- and megakaryopoesis). Decreased bone marrow cellularity is commonly found after CAR-T treatment as discussed in the manuscript and the underlying mechanism for the bone marrow hypocellularity in this regard is not well defined so far. Evaluation of complement levels is a very good suggestions, which we have not done in these two patients. We have added this to the discussion, so that it can be analyzed in the future if we or others observe this constellation again (“We want to point out important limitations of our case study since we did not look for signs of complement activation (i.e. C3 and C4 levels) and did not test for paroxysmal nocturnal hemoglobulinuria as possible differential diagnosis. We would recommend testing for this if future similar cases occur.”, line 228-231). And: “Decreased bone marrow cellularity is commonly described after CAR-T treatment, and the reduction in thrombo- and granuopoiesis is in line with previous reports about prolonged cytopenias after CAR-T treatment. However, erythropoietic precursors were increased in our cases, which is in accordance with the observed peripheral de-struction of RBCs. In one patient we had the possibility to measure CAR-T levels in the peripheral blood at this time. Interestingly, we observed a decline in peripheral CAR T-cell levels over time coinciding with resolution of hemolysis. Given the timely corre-lation with the CAR-T treatment, we speculated whether there was an association with the hemolytic anemia.” (line 184-191)

  1. Authors suggest that release of cytotoxic granules (presumably by CAR-T cells) would cause hemolysis as “bystander effect”. To me this explanation seems illogical as patients have reached remission and also the amount of the target of CAR-T cells (CD19) has significantly diminished at the time when hemolysis was detected. Why this “bystander effect” does not manifest earlier when CAR-T activity is at its highest and CAR-T cells are releasing cytotoxic granules most?

Response: We thank the reviewer for this valuable input, which we can completely understand. At the moment we can only speculate on the underlying mechanism. It seems that there is direct damage to red blood cells in the circulation, which is not mediated by an immune mechanism. This destruction stops, when CAR-T levels decline (which clearly is only an association and those not prove causality). Probably, destruction of RBCs was caused by cytotoxic granules, which are released upon CAR T-cell binding to the target cell as a sort of “bystander effect” but we can only speculate about this. We have added this information as a point of discussion to text (Line 193-202)

“Probably, destruction of RBCs was caused by cytotoxic granules, which are released upon CAR T-cell binding to the target cell as a sort of “bystander effect” but we can only speculate about this. We argue that most of CAR-T cell mediated lymphoma cell de-struction will take place at the tumor site shortly after infusion (in our case lymph nodes, spleen, liver and bone marrow) and not the peripheral blood. When patients start to reconstitute some CD19 positive healthy B-cells in the blood after some weeks to months, they will eventually be recognized by CAR-T cells and consecutively be eliminated. Probably, some RBCs will get destroyed as a part of this process. In line with this postulated mechanism we observed an incline in CD19 levels exactly during the time of hemolysis and declining CAR T-cells.”

  1. There is a typo in line 37. “Pediatric B-acute lymphoblastic anemia” should be “pediatric acute B-cell lymphoblastic leukemia”.

Response: This has been corrected. (line 37)

Reviewer 2 Report

The Authors describe to cases of pancytopenia and DAT negative hemolytic anemia following CART therapy. Cytopenias spontaneously ameliorated over time and in one patient this was concomitant to CART cells reduction in the peripheral blood. 

Although there is growing interest in early and late onset cytopenias following CART therapy, this report lacks important information that may improve pathogenic insights and how these patients are handled.

  1. how many patients have the Authors treated with CART cells at their centre?
  2. how many of them did develop a cytopenic episode? how many thrombocytopenia, anemia, neutropenia, or bi-, pancytopenia?
  3. home many patients developed hemolytic features?
  4. how is the timing of cytopenia development in respect to CART cell infusion?
  5. are cytopenias correlated to CRS or other side effects?
  6. Have these cytopenias ever required treatment? (steroids, IVIG...?)
  7. All these information would help to speculate about the frequency of this side effect and on its pathogenesis, and I feel that the Authors may collect them and discuss them.
  8. In the case reports, next to the graphs (figure 1), a table showing the dynamics of inflammatory markers could be of aid (ESR, PCR, ferritin, ...) to better understand the physiopathology of these cytopenias.
  9. Was paroxysmal nocturnal hemoglobinuria ruled out (cause of DAT neg hemolytic anemia) ?
  10. How were complement levels in these patients (C3, C4..)?
  11. How was the direct anti-globulin test performed?
  12. Have endogenous erythropoietin levels been tested?
  13. When CMV and EBV are cited, Parvovirus should also be exlcuded
  14. It may also be worth mentioning the eclusion of HLH (along with TMA) and the following could be cited "PMC7923749
  15. Finally a table and brief description of the available literature on cytopenias following CART therapy and how to manage it would be highly welcome. (See also registrative trials and further studies).

Author Response

Dear reviewer, 

We would like to thank the reviewer the very valuable input and suggestions as well as interest in CAR T-cell treatment at our center. Given multiple great comments, we believe this manuscript has greatly improved. We would like to add that large areas of the discussion have been restructured thanks to several suggestions, and we consider this to contribute significantly to the improvements of our manuscript. In the following lines, we would like to address all points.

  1. how many patients have the Authors treated with CART cells at their centre?

Response: At our center, 9 patients have undergone CAR T-cell therapy over the last two years. Of note, one of the patients in this case report was referred to another center. Apart from that, however, the patient was treated at our center. We added “At our center, 9 patients have been treated with CAR T-cells so far, but the phenomenon of hemolytic anemia was not observed in any of the other patients..” in line 170-172 to the discussion.

  1. how many of them did develop a cytopenic episode? how many thrombocytopenia, anemia, neutropenia, or bi-, pancytopenia?

Response: Virtually all patients developed some kind of cytopenia, some of which were relatively prolonged. Our experience regarding prolonged cytopenias thus far is comparable to published data from the pivotal clinical trials, besides from the two cases which we reported here.

  1. how many patients developed hemolytic features?

Response: These were the only two patients treated/followed up at our center that developed hemolysis. Inaddition to that, we did not find any cases reporting on a relation between hemolytic anemia and CAR T-cell therapy so far.

  1. how is the timing of cytopenia development in respect to CART cell infusion?

Response: Almost all patients will develop cytopenia within the first days after CAR T-cell treatment. This is transient and most often interpreted as an effect of the lymphodepleting therapy and will resolve after 28 days in most patients. Prolonged G3/4 cytopenias (i.e. lasting longer than 28 days by definition) “However, prolonged (i.e. lasting longer than 28 days) grade 3-4 cytopenias are reported in 32% and 48%  of patients in the ZUMA-16 and JULIET7 trials, respectively and in Twenty six percent of patients in the ZUMA-23 trial had grade 3-4 cytopenias after 90 days (table 1)We added this information into the new table 1 at the end of the text and referenced it from the introduction in line 44-47.

  1. are cytopenias correlated to CRS or other side effects?

Response: In fact, current evidence suggests a correlation of prolonged cytopenia with severity of CRS and neurotoxicity. We have added this information to the text: “Current literature reports an association of prolonged cytopenia with CRS or ICANS > grade 3 as well as baseline cytopenia, and inflammatory markers.” (line 51-52)

  1. Have these cytopenias ever required treatment? (steroids, IVIG...?)

Response: Cytopenias were treated supportively with red blood cell or platetet transfusions and growth factor support. Since we have excluded an immune-mediated effect (negative DAT test), we did not administer steroids or IVIG for the anemia in these cases.

  1. All these information would help to speculate about the frequency of this side effect and on its pathogenesis, and I feel that the Authors may collect them and discuss them.

Response. We thank the reviewer for these suggestions. Questions 1-6 are somewhat interrelated. The cohort of patients treated at our center is continuously growing and data is collected prospectively. The small number of patients, however, does not allow us to draw valid conclusions in this regard, but we included the number of patients into the discussion (“At our center, 9 patients have been treated with CAR T-cells so far, but the phenomenon of hemolytic anemia was not observed in any of the other patients..”, lines 156-158).

  1. In the case reports, next to the graphs (figure 1), a table showing the dynamics of inflammatory markers could be of aid (ESR, PCR, ferritin, ...) to better understand the physiopathology of these cytopenias.

Response: We have included the following information on this topic: Ferritin levels were constantly elevated after CAR T-cell therapy and found to be between 1000 and 2000 µg/L. CRP levels were normal or occasionally slightly elevated, and ESR was not determined.Ferritin levels were constantly elevated between 1400 and 2200 µg/dL after CAR T-cell treatment and CRP was normal to minimally elevated up to 1,2 mg/dL” (line 97-99). “Also in this patient, ferritin levels were elevated (between 1000 and 1700 µg/dL) after CAR T-cell therapy with normal to slightly elevated CRP up to 1,1 mg/d)” (line 137-139). We would prefer to mention this briefly in the results only, as we did unfortunately not see a clear dynamic in these parameters.

  1. Was paroxysmal nocturnal hemoglobinuria ruled out (cause of DAT neg hemolytic anemia) ?

Response: We thank the reviewer for this valuable input. We did not test for PNH, however, we think that this diagnosis is unlikely because of the spontaneous resolution of the hemolysis without treatment. It is now mentioned as a limitation of our study: “We want to point out important limitations of our case study since we did not look for signs of complement activation (i.e. C3 and C4 levels) and did not test for paroxysmal nocturnal hemoglobulinuria as possible differential diagnosis. We would recommend testing for this if future similar cases occur.” 228-231

  1. How were complement levels in these patients (C3, C4..)?

Response: We did not assess complement levels at that time, because the observed changes came rather surprisingly to us. We completely agree with the reviewer, that this should be assessed. We have now mentioned this aspect as a limitation of our report in the discussion. (““We want to point out important limitations of our case study since we did not look for signs of complement activation (i.e. C3 and C4 levels) and did not test for paroxysmal nocturnal hemoglobulinuria as possible differential diagnosis. We would recommend testing for this if future similar cases occur. Line 228-231)

  1. How was the direct anti-globulin test performed?

Response: DAT tests are performed by our colleagues at the Department of Transfusion Medicine. At our center, we use ID- Cards DC Screening I, consisting of five different monospecific AHG reagents (Anti- IgG, -IgA, - IgM, - C3c, C3d), suspended in gel.

  1. Have endogenous erythropoietin levels been tested?

Response: No, we have not determined EPO levels, as we attributed the anemia to hemolysis (as outlined in the manuscript) and did not observe a decreased production of red blood cells as one would expect with decreased EPO levels

  1. When CMV and EBV are cited, Parvovirus should also be exlcuded

Response: We thank the reviewer for this input. Parvovirus infection was excluded in patient B (“EBV,CMV and parvovirus”, line 134) but not in patient A.  We agree that parvovirus infection should have been ruled out in both patients.

  1. It may also be worth mentioning the exlusion of HLH (along with TMA) and the following could be cited "PMC7923749

Response: We value this contribution. Although we did not deterimine CD25/IL2R alpha in our patients, the diagnostic criteria for HLH would not be fulfilled in our patients (max 3-4 points, 5 required for diagnosis). This was included into the discussion as: “Of note, bone marrow did not show any signs for hemophagocytosis and patients did not fulfill HLH-2004 diagnostic criteria for hemophagocytic lymphohistiocytosis..”, line 180-182

  1. Finally a table and brief description of the available literature on cytopenias following CART therapy and how to manage it would be highly welcome. (See also registrative trials and further studies).

Response: We have added a table describing the percentage of late cytopenias occurring in major clinical trials. (Table 1, line 241-243) Regarding management, we included a short paragraph discussing the fact that supportive management is currently recommended (also answers number 7): “As reviewed by Penack and Koenecke in 2020, prolonged cytopenia is still of unclear origin in most patients. Regarding management recommendations, Neelapu list the possibility of growth factor support and transfusions as necessary options and advice antibiotic/antimycrobial prophylaxis for prolonged neutropenia. The current EBMT guidelines recommend routine monitoring of complete blood count 28 days after therapy, but do not specifically advise on monitoring for hemolysis and do not give specific recommendations regarding management.” (line 233-239)

Round 2

Reviewer 1 Report

Comments to the authors

The authors have adequately addressed my comments and I think that the manuscript is now publishable in International Journal of Molecular Sciences